# Target Score—A Proteomics Data Selection Tool Applied to Esophageal Cancer Identifies GLUT1-Sialyl Tn Glycoforms as Biomarkers of Cancer Aggressiveness

**DOI:** 10.3390/ijms22041664

**Published:** 2021-02-07

**Authors:** Sofia Cotton, Dylan Ferreira, Janine Soares, Andreia Peixoto, Marta Relvas-Santos, Rita Azevedo, Paulina Piairo, Lorena Diéguez, Carlos Palmeira, Luís Lima, André M. N. Silva, Lúcio Lara Santos, José Alexandre Ferreira

**Affiliations:** 1Experimental Pathology and Therapeutics Group, IPO Porto Research Center (CI-IPOP), Portuguese Oncology Institute (IPO Porto), 4200-072 Porto, Portugal; sofia.ribeiro.cotton@ipoporto.min-saude.pt (S.C.); dylan.ferreira@hotmail.com (D.F.); janinepaivasoares@gmail.com (J.S.); andreia.peixoto@ipoporto.min-saude.pt (A.P.); marta_frds@hotmail.com (M.R.-S.); carlospalmeira@ipoporto.min-saude.pt (C.P.); luis.carlos.lima@ipoporto.min-saude.pt (L.L.); llarasantos@gmail.com (L.L.S.); 2Institute of Biomedical Sciences Abel Salazar (ICBAS), University of Porto, 4050-313 Porto, Portugal; 3Institute for Research and Innovation in Health (i3S), University of Porto, 4200-135 Porto, Portugal; 4Institute for Biomedical Engineering (INEB), 4200-135 Porto, Portugal; 5QOPNA/LAQV, Department of Chemistry, Campus Universitário de Santiago, University of Aveiro, 3810-193 Aveiro, Portugal; 6REQUIMTE-LAQV, Department of Chemistry and Biochemistry, Faculty of Sciences, University of Porto, 4169-007 Porto, Portugal; andre.silva@fc.up.pt; 7Institute of Biomedicine, University of Turku, FI-20014 Turku, Finland; rita.pereiraazevedo@utu.fi; 8Medical Devices Research Group, International Iberian Nanotechnology Laboratory (INL), 4715-330 Braga, Portugal; paulina.piairo@inl.int (P.P.); lorena.dieguez@inl.int (L.D.); 9Department of Immunology, Portuguese Institute of Oncology of Porto, 4200-072 Porto, Portugal; 10Health Science Faculty, University of Fernando Pessoa, 4249-004 Porto, Portugal; 11Porto Comprehensive Cancer Center (P.ccc), 4200-072 Porto, Portugal; 12Department of Surgical Oncology, Portuguese Institute of Oncology, 4200-072 Porto, Portugal

**Keywords:** cancer biomarkers, circulating tumors cells, esophageal cancer, glycomics, glycoproteomics, bioinformatics

## Abstract

Esophageal cancer (EC) is a life-threatening disease, demanding the discovery of new biomarkers and molecular targets for precision oncology. Aberrantly glycosylated proteins hold tremendous potential towards this objective. In the current study, a series of esophageal squamous cell carcinomas (ESCC) and EC-derived circulating tumor cells (CTCs) were screened by immunoassays for the sialyl-Tn (STn) antigen, a glycan rarely expressed in healthy tissues and widely observed in aggressive gastrointestinal cancers. An ESCC cell model was glycoengineered to express STn and characterized in relation to cell proliferation and invasion in vitro. STn was found to be widely present in ESCC (70% of tumors) and in CTCs in 20% of patients, being associated with general recurrence and reduced survival. Furthermore, STn expression in ESCC cells increased invasion in vitro, while reducing cancer cells proliferation. In parallel, an ESCC mass spectrometry-based proteomics dataset, obtained from the PRIDE database, was comprehensively interrogated for abnormally glycosylated proteins. Data integration with the Target Score, an algorithm developed in-house, pinpointed the glucose transporter type 1 (GLUT1) as a biomarker of poor prognosis. GLUT1-STn glycoproteoforms were latter identified in tumor tissues in patients facing worst prognosis. Furthermore, healthy human tissues analysis suggested that STn glycosylation provided cancer specificity to GLUT1. In conclusion, STn is a biomarker of worst prognosis in EC and GLUT1-STn glycoforms may be used to increase its specificity on the stratification and targeting of aggressive ESCC forms.

## 1. Introduction

Esophageal cancer (EC) poses as a major health risk due to increasing incidence, late diagnosis and poor prognosis [1]. It ranks seventh in terms of incidence and sixth in terms of cancer mortality worldwide, with over 500,000 new cases a year and 5-year overall survival rate of approximately 20% [2]. Esophageal squamous cell carcinoma (ESCC) arising from the non-keratinized stratified squamous epithelium is the most prevalent histological subtype worldwide (60–70%) and occurs more often in the upper and mid-esophagus [3]. Although surgery, chemo- and radiotherapy are widely used to manage ESCC patients, responses to treatment remain poor and encompass relevant health side effects [4], urging the introduction of novel and more effective therapeutic modalities to avoid tumor relapse and rapid disease dissemination. However, high inter- and intra-tumoral molecular heterogeneity poses a significant hurdle for the identification of cancer-specific molecular targets, decisively delaying accurate patient stratification and effective targeted therapeutics.

In recent years, many studies have comprehensively interrogated the proteome of EC in search for biomarkers to aid disease management at different levels, including diagnosis, patient stratification, prognosis and responses to treatment [5,6,7]. Despite promising, conventional proteomics approaches on EC are yet to deliver targetable molecular signatures due to insufficient cancer specificity. Addressing alterations in the glycosylation of membrane proteins holds a great potential towards this objective [8]. In fact, over the past years, we and other research groups have demonstrated that changes in the structure and distribution of glycans in proteins at the surface of cancer cells may provide the necessary bispecificity towards unique cancer molecular signatures [8,9]. Namely, we have identified MUC16, CD44 and nucleolin glycoforms holding potential for more accurate patient stratification and treatment follow-ups in different types of tumors [9,10,11]. We have also used patient’s autoantibodies to demonstrate that changes in EC glycosylation could generate cancer glyconeoantigens, reinforcing the enormous potential of glycoproteomics for biomarker discovery [12]. However, pinpointing clinically relevant molecular signatures from big datasets remains a daunting enterprise, requiring dedicate bioinformatics tools. Recently, we have addressed this aspect by proposing a simple algorithm, termed Target Score, that exploits clinicopathological data deposited in the Human Protein Atlas to extract cancer biomarkers from glycoproteomics experiments [9]. Our approach was specifically designed to rank proteins according to its prognosis value and targetability, favoring proteins located at the cell surface, showing high abundance in tumors in relation to healthy human organs. Furthermore, it significantly penalized proteins expressed in reproductive, immune and nervous systems, allowing to choose molecules posing minimum probability of significant off-target effects. Target Score-assisted glycoproteomics in gastric cancer led to the identification of nucleolin glycofoms at the cell surface associated with worst prognosis and metastases development, which contrasted with its typical nuclear location in healthy tissues. The cancer-specific nature of this finding elegantly portrayed the potential of Target Score algorithm for identification of unexpected glycosignatures and set a biomarker discovery roadmap that can now be translated and adapted to other human tumors, including EC [9].

Glycobiomarkers discovery often requires the adoption of targeted glycoproteomics approaches focusing on specific alterations in protein glycosylation associated with cancer [9,10,13]. In this context, digestive tract tumors have been long known to express membrane proteins carrying abnormal glycosylation patterns as post-translational modifications [8,9,14]. Namely, these tumors often overexpress sialyl-Tn (STn), which results from a premature stop in *O*-glycans elongation by sialylation of the Tn antigen [8]. The STn antigen is highly expressed by more aggressive and advanced gastric and colorectal tumors, being generally associated with poor prognosis [8]. Moreover, it plays a relevant functional role in cancer, namely chemoresistance [15], cell invasion [16,17] and immune escape [18,19], decisively contributing to disease progression. The STn antigen is also rarely observed in most healthy tissues, except for low expression in cells specialized in mucin secretion facing the lumen of the gastrointestinal and respiratory tracts [20]. As such, targeting STn and STn-expressing glycoproteins constitutes a relevant approach towards identifying cancer-specific glycobiomarkers in EC. However, little is known about STn expression patterns and its clinical relevance in these tumors. Therefore, the present work devotes to providing a clinical and functional context for STn in ESCC, which constitute the bulk of EC. Furthermore, it comprehensively interrogates a glycoproteomics dataset for STn-expressing glycoproteins. Potentially targetable biomarkers arising from this study were subsequently validated in EC and healthy human tissues foreseeing the molecular rationale for precision oncology.

## 2. Results and Discussion

The present study builds on the hypothesis that the STn antigen may be explored towards the objective of more precise patient stratification and targeting ESCC, given its functional role in the disease and frequent association with poor prognosis in many solid tumors [8,17,18,21]. Therefore, we first focused in setting the clinical context for STn expression using ESCC tissues of different clinicopathological natures and exploiting its functional role through a glycoengineered cell model. The second part of the study is concerned with the identification of cancer differently expressed proteins which bear the STn antigen, using database information and subsequent validation in tumor tissues.

### 2.1. STn Antigen in ESCC Primary Tumours, Metastases and CTCs

Little is known about the expression of STn antigen in ESCC, which constitutes the main histopathological group within EC. Therefore, we have evaluated its expression in a retrospective series of 48 cancer tissues reflecting different stages and grades of the disease (Table 1). Approximately 70% of tumor tissues were positive for the antigen, with a marked membrane expression mainly in tumor cells. Notably, cancer cells expressing this antigen were found in scattered clusters that, for most cases, did not exceed 20% of the tumor area (Figure 1A). We could also find STn antigens in adjacent lymph node metastases, suggesting a role in disease dissemination via lymphatic vessels, which warrants future confirmation. The presence of STn was further confirmed by the lack of immunoreactivity for B72.3 plus CC49 antibodies in tissue sections incubated with neuraminidase prior to analysis. Finally, we have used innovative microfluidics devices to capture CTCs from the blood of ESCC patients and assess STn expression in situ. Briefly, peripheral blood was prospectively collected from 10 patients without prior treatment history (Table 2) and processed using the microfluidic chips. Cancer cells with larger dimensions and less deformable were entrapped in the filter area with minimal contamination from smaller and deformable blood cells [22]. Based on our previous observations, the absence of CD45 and presence of pan-CK and/or STn were considered for definitive tumor cell identification [21,22]. Even though early stage ESCC, which constitutes the bulk of the prospective series, are known to shed modest numbers of cancer cells into hematogenous circulation [23], we were able to detect STn positive CTCs (DAPI^+^, STn^+^, pan-CK^−^, CD45^−^) in 20% of the patients (Figure 1B). Interestingly, STn expressing CTCs did not present pan-CKs expression typical of cells of epithelial nature, suggesting a mesenchymal-like phenotype which also warrants confirmation. Although the low number of cases did not allow the establishment of associations with clinicopathological features, the discovery of STn in ESCC CTCs supports the need for more in-depth investigation on this subject. In fact, the detection of STn antigens in lymph node metastases and CTCs supports a role in disease dissemination, which may be of great value for improved liquid biopsies, as previously suggested for other solid tumors [21,22]. Notably, the STn antigen was also detected in the histologically normal esophageal mucosa adjacent to tumor tissue, mainly in the differentiating cells of the mucosa facing the esophagus lumen (Figure 1A). Notwithstanding, mucosal cells showed considerably lower STn expression than tumor cells and of predominantly cytoplasmatic nature, demanding a comprehensive characterization of the STn-glycoproteome for improved cancer specificity.

Finally, we evaluated possible links between STn expression in tumors and clinically relevant clinicopathological variables as stage, TNM staging, keratinization degree, differentiation degree, extra-tumoral growth, lympho-vascular permeation and perineural permeation. No evident associations were established (Table 3), reflecting the prevalent nature of this antigen across the disease. However, a statistically significant association was observed regarding recurrence (Figure 1C) and decreased disease-free survival (Figure 1D), supporting a role in tumor aggressiveness.

### 2.2. Generation of a STn ECSS Cell Line and Functional Implications

Analysis of clinical samples strongly supported a link between the STn antigen and ESCC progression and dissemination, as previously observed for other solid tumors [17,24]. To gain more insights on this matter, we elected the Kyse-30 cell line as in vitro model to explore the functional impact of STn in ESCC. We started by characterizing the cellular *O*-glycome, using a benzyl-GalNAc residue, which mimics the Tn antigen, the first sugar in *O*-glycans biosynthesis [25]. The Tn mimetic was added to the culture medium and used by cancer cells glycosylation machinery as a scaffold for further elongation. After processing across the secretory pathways, more extended benzylated glycans reflecting the cells *O*-glycome were secreted and easily recovered from the culture media by C18 reverse phase affinity, permethylated and analyzed by nanoLC-ESI-MS/MS. According to Figure 2A, Kyse-30 mainly expressed extended core 2 *O*-glycans (*m*/*z* 1021.53, 1195.62, 1382.71, 1470.76) as well as T (*m*/*z* 572.31) and sialyl-T antigens (*m*/*z* 933.48). To a lesser extent, this cell line also expressed core 3 *O*-glycan (*m*/*z* 613.33). Moreover, mass spectrometry did not shown signs of the STn antigen (Figure 2A). Collectively, we have demonstrated that Kyse-30 cells presented a glycome marked mainly by extended core 2 *O*-glycans, therefore not reflecting the presence of STn exhibited by some subpopulations of cancer cells in human ESCC tumors. Notably, the absence of short-chain *O*-glycans such as the STn antigen in Kyse-30 is consistent with many other reports for human cancer cell lines grown in vitro, denoting a strong dependence on yet incompletely determined microenvironmental features [16,26]. We believe that the identification of the microenvironmental elements underlying *O*-glycans antagonization in vivo will be critical to provide a clear context for clinical interventions and should be addressed in a close future [27].

We then induced the stable overexpression of ST6GalNAc I, responsible by the *O*-6 sialylation of the Tn antigen originating STn epitopes, originating the Kyse-30 *ST6GALNAC1* KI cell line. As shown in Figure 2A, Kyse-30 *ST6GALNAC1* KI cells gained the capacity to express the STn, whereas the antigen was not observed in mock cells. This was obtained without inducing major glycome remodeling in comparison to wild type and mock cells (Figure 2A). Flow cytometry analysis confirmed that Kyse-30 cells did not express the STn antigen, whereas Kyse-30 *ST6GALNAC1* KI exhibited high levels of this antigen in approximately 90% of cells (Figure 2B). Subsequent functional studies showed that STn overexpression was accompanied by a striking decrease in cell proliferation (55 and 70% in relation to mock and wild type cells; Figure 2C), suggesting that STn expression may contribute to a quasi-quiescent state, which is frequently observed in more aggressive cancer cell phenotypes. Concomitantly, STn-expressing cells significantly increased their invasive capacity in vitro compared to both mock and wild type control cells (Figure 2D), as previously observed for other glycoengineered cancer cell models of different origins (breast, gastric, bladder) [24,28,29]. Collectively, these findings reinforce the close link between STn expression and ESCC aggressiveness, as previously observed for other types of solid tumors [8,17,24]. Namely, it is well described that the substitution of elongated glycans by shorter sialylated antigens, such as STn, promotes a profound remodeling of the plasma membrane glycoproteome, triggering important oncogenic signaling pathways involved in cell proliferation [30,31,32]. Moreover, it changes cell-cell and cell-matrix adhesion interfaces, facilitating cell motility [16,27]. Nevertheless, the exact molecular mechanism governing the functional alterations resulting from STn overexpression in Kyse-30 cells remains unknown and should be subject to future investigation, envisaging a more detailed understanding on the biological role of STn in esophageal cancer.

### 2.3. Identification of STn Modified Proteins

Immunohistochemistry analysis showed that both ESCC and histologically normal esophageal tissues express STn antigens. Even though with distinct intensities and tissue-specific patterns, these observations challenge the cancer specificity of the STn antigen alone. This led us to query if investigating the STn-associated glycoproteome can contribute to increase cancer specificity. To address this objective, we opted for a bioinformatics approach comprehending four main steps, summarized in Figure 3A. Namely, (i) identification of ESCC proteomics raw data deposited in relevant primary databases, which could serve as starting point for our research; (ii) identification of membrane proteins that, due to their cell surface nature, may be easily targetable by antibodies and other ligands; (iii) sorting of potentially targetable glycoproteins based on their overexpression in cancer in relation to healthy tissues, with the assistance of the target score algorithm; (iv) identification of STn-glycosites in top ranked proteins holding potential for future design of targeted therapeutics.

After an initial screening of the PRIDE database and despite the significant number of proteomics studies on EC, we have identified only one ESCC proteomics dataset freely available to the scientific community at the time of this study. It concerned a pool of tumor proteins from 6 patients, however, without a clear identification of the tumors clinicopathological natures. Despite the preliminary nature of the study, we decided to use it as starting point to portrait the potential of our approach for glycobiomarkers discovery. As a general approach, we re-analyzed data against a database composed of proteins with Gene Ontology (GO) terms for the plasma membrane (Figure 3A), leading to the identification of 773 and 795 plasma membrane proteins for RPS and SCX pre-purification methods, respectively (Appendix A). To reduce the dataset dimension to very high confidence identifications for downstream analysis, we have focused on the top 250 glycoproteins from each dataset (according to the posterior error probability-PEP score). Notably, we have identified 213 glycoproteins that were common to the RPS and SCX methods (Appendix A). This list was after investigated with the Target Score algorithm, a tool designed by our group to extract clinically relevant information in a comprehensive manner from large protein lists retrieved from proteomics studies. This algorithm uses protein expression data from the Human Protein Atlas to pinpoint potentially targetable glycobiomarkers by attributing higher scores to proteins expressed in higher abundance at the cell surface in tumors, while penalizing high expression in healthy tissues. Finally, it also potentiates associations with worst cancer prognosis, which help set the pace for addressing more aggressive tumors. However, there is still limited data on ESCC in comparison to other solid tumors. To address this limitation, we used head-and-neck squamous cell carcinomas as models based on similar histology, etiology and molecular features [33,34,35]. As shown in Figure 3B, SLC2A1, also known as glucose transporter type 1 (GLUT1), was retrieved with the highest target score (Appendix A). GLUT1 has been previously found to be overexpressed in ESCC in response to metabolic adaptations to rapid tumor growth and hypoxia [36,37]. Moreover, GLUT1 expression has been associated with hematogenous recurrence [38] and considered a surrogate marker of decreased response to cisplatin [39] and worst prognosis following potentially curative surgical resection [40]. Collectively, these observations support a key role for this protein in esophageal cancer, supporting its high target score. However, GLUT1 as all identified cell membrane proteins presents some degree of expression in healthy tissues, limiting their utilization for precise cancer targeting (Appendix A). Therefore, we evaluated the possibility of obtaining a higher degree of cancer specificity when considering the GLUT1-STn glycoproteoforms. We started by re-addressing the mass spectrometry datasets with Byonic, a dedicated protein assignment software which facilitates the assignment of post-translational modifications in glycopeptides [41]. Only glycopeptide spectra presenting typical GalNAc and sialic acid oxonium ions derived from STn were considered for glycosites assignment, as exemplified by the HCD-MS/MS spectra on Figure 3C. Even though the original proteomics experiment was not designed for glycopeptides identification, we were able to confirm 18 STn-glycosites GLUT1, highlighted in Figure 3D and Appendix A. These, however, should now be carefully validated in individual tumor samples. Despite the exploratory nature of this approach, we were able to identify the GLUT1-STn glycoproteoforms as a potential ESCC specific molecular signature.

### 2.4. GLUT1 in ESCC

The suggestion of GLUT1-STn as a specific marker for ESCC prompted a reinvestigation of our retrospective tumor series for this glycoprotein. We found GLUT1 widely expressed across tumor tissues at the cell surface of cancer cells, even though with variable intensity and extension between individual cases (Table 4; Figure 4A). GLUT1 could also be observed in all metastasized lymph nodes. The staining extension exceeded 40% of the tumor area for all studied cases, without a defined histological pattern. However, staining intensities varied from weak to very strong. Tumor-associated immune infiltrates also presented strong GLUT1 membrane expression. In addition, the adjacent but histologically healthy esophageal mucosa exhibited a strong membrane staining restricted to cells of the basal layer of the squamous epithelium (Figure 4A). Although lacking tumor specificity, high GLUT1 levels, characterized by both high extension and high staining intensity, showed a trend association with the presence of metastases and associated with distant recurrence (*p* = 0.026, Table 4), reinforcing observations from other authors linking this protein to metastasis [42,43,44]. Building on previous insights from glycoproteomics analysis, we were then led to investigate the presence of GLUT1-STn in individual tumors.

A comparative analysis of STn and GLUT1 on consecutive immunohistochemically labeled tissue sections revealed the co-expression of GLUT1 and STn in the same tumor areas as well as in the metastases (Figure 4A). Immunofluorescence double staining for these antigens in the same tumor section confirmed that both antigens were in the same cancer cells, irrespectively of the stage of the tumor (Figure 4B), strongly supporting the existence of GLUT1-STn, which was later confirmed by western blot (Figure 4C). We started by screening whole protein extracts isolated from FFPE tissues showing high GLUT1 and STn by western blot with no success (Figure 4C), suggesting low abundance of membrane proteins [9]. To overcome this limitation, GLUT1 was immunoprecipitated from tumors showing high GLUT1 and STn co-expression in the same area and immunoblotted for both GLUT1 and STn. We started by noting that all blots presented a strong band above 250 kDa that we believe to be mucins known to carry the STn antigen (Figure 4C). These glycoproteins are abundantly present along the gastrointestinal tract and overexpressed in ESCC and may often co-precipitate in immunoaffinity assays. Besides these bands, GLUT1 immunoprecipitates showed a major band above 75 kDa, mostly likely corresponding to highly glycosylated forms of the glycoprotein and a faint band at approximately 50 kDa that was also observed in the anti-STn blots and most likely belongs to GLUT1-STn glycoproteforms (Figure 4C). Notably, none of these bands were present in isotype negative controls, reinforcing previous assignments. Moreover, the lower abundance of GLUT1-STn band in relation to the most expressed form of GLUT1 was consistent with the focal expression presented by STn in comparison to the diffuse nature of GLUT1 in ESCC. Notably, according to Uniprot (https://www.uniprot.org/ (accessed on 1 January 2021)), GLUT1′s canonical form has approximately 54 kDa, which may be significantly increased by post-translational modifications explaining the band at 75 kDa. In fact, bioinformatics predictions showed for this protein two potential *N*-glycosites (according to NetNGlyc; http://www.cbs.dtu.dk/services/NetNGlyc/ (accessed on 1 January 2021) [45]) and up to 14 predicted *O*-glycosites, depending on available polypeptide *N*-acetylgalactosaminyltransferases (according to ISOGlyP; https://isoglyp.utep.edu/ (accessed on 1 January 2021) [46]). Replacement of extended by shorter glycoforms such as the STn antigen may explain the existence of a lower molecular weight GLUT1 band. The fact that this band was found slightly below the estimated for the canonical form of the protein may derived from alterations in electrophoretic mobility, which are common upon changes in glycosylation, namely sialic acids content [47,48]. Furthermore, it should be noted that samples were obtained from FFPE tissues and it is well known that pre-analytical factors (both the fixation and the solubilization protocols) may introduce protein modifications, such as amino acid side chain PTMs or protein backbone hydrolysis, which could result in the alteration of the protein molecular weight and electrophoretic migration [49,50,51,52]. These modifications, together with difficulties in predicting the MW shift induced by glycosylation make it difficult to interpret WB singly on the MW predicted from protein amino acid sequence. Finally, we cannot exclude the expression of shorter GLUT1 proteoforms, as frequently observed for other proteins in cancer [53] and have been predicted by Uniprot. Faced with these preliminary observations, we are engaging in more comprehensive glycoproteogenomics studies to fully characterize the nature of GLUT1 glycoforms in ESSC. Nevertheless, collectively, western blot supports the existence of GLUT1-STn glycoproteoforms in ESCC suggested by immunoassay, as previously hinted by our in silico approach. Interestingly, STn and GLUT1 were not co-expressed in the same areas of the healthy esophagus (Figure 4A).

Having validated the presence of GLUT1-STn in ESCC, we then devoted to seeking potential associations with clinical variables. We found that patients with STn negative tumors and expressing low levels of GLUT1 exhibited longer disease-free survival compared to patients with tumors presenting other glycophenotypes (*p* = 0.027, log-rank; Figure 4D). In addition, STn positive tumors with high GLUT1 favored worse patient survival, reinforcing the close link between the GLUT1-STn molecular signatures and cancer aggressiveness. According to the Cox Regression model, the GLUT1-STn overexpression phenotype in ESCC constitutes an independent prognosis factor regarding disease-free survival.

In summary, we have presented strong evidence that the GLUT1-STn glycophenotype is characteristic of more aggressive ESCC, associating with decreased survival, which may be helpful for patient stratification. Moreover, it could not be observed in healthy esophageal tissue, holding high cancer specificity.

### 2.5. STn and GLUT1 in Health Tissues

To address the cancer-specific dimension of GLUT1-STn, we started by comparing GLUT1 and ST6GalNAc I expressions in a wide variety of healthy human tissues, exploring the Human Protein Atlas (Figure 5A). ST6GalNAc I, the glycosyltransferase responsible by the *O*-6 sialylation of the Tn antigen, was used as surrogate for STn biosynthesis potential, overcoming the lack of systematized information on *O*-glycans expression in human tissues. However, we note that ST6GalNAc I detection cannot be regarded as definitive proof for presence of STn, given the complex nature of events governing *O*-glycosylation. According to Figure 5A, GLUT1 is present in few healthy tissues compared to ST6GalNAc I, namely the placenta, kidney, skin, tonsil and the cerebral cortex. Interestingly, the Human Protein Atlas did not report GLUT1 in the esophagus, which contrasts with our experimental data demonstrating low to moderate expressions in the basal layer of the squamous epithelium. These discrepant observations may result from differences in the specificity of the used antibodies and support more detailed reinvestigation of GLUT1 in healthy tissues by quantitative high-throughput technologies such as mass spectrometry. On the other hand, ST6GalNAc I was mostly found across the gastrointestinal and respiratory tracts, female and male organs and the skin, in agreement with previous reports for STn [20]. Interestingly, the organs that presented GLUT1 were also positive for ST6GalNAc I. However, a closer look showed significant differentiation at the cell level in the placenta and the cerebral cortex (Figure 5B), as previously observed by us for the esophagus (Figure 4A). On the other hand, GLUT1 and ST6GalNAc I were observed in the same type of cells in the kidney and skin. The subsequent screening of healthy kidney tissue sections by immunohistochemistry for GLUT1 and STn supported the presence of STn in tubular cells of the kidney but not GLUT1 (Figure 5C) as suggested by the Human Protein Atlas, which possibly translates differences in antibody affinities. On the other hand, we could find GLUT1 in the healthy skin but not STn, vowing for the absence of GLUT1-STn glycoforms (Figure 5C). We then expanded this study to a wider array of healthy tissues consisting of thyroid, liver, gallbladder, testis, lung, stomach, pancreas and colon samples. STn presented a restricted distribution in these tissues, mostly circumscribed to the cytoplasm of the parietal and goblet cells of the respiratory and gastrointestinal tracts as well as in sertoli cells of the testicles, thus in agreement with the presence of ST6GalNAc I (Figure 5A). Nevertheless, the intensity of expression was much lower in comparison to esophageal tumors. On the other hand, using this approach GLUT1 expression in healthy tissues was only detectable in the cell membrane of immune cells in stomach, liver, colon, lung and pancreatic tissues that did not express STn but not in the cells of the targeted organs, again in agreement with Figure 5A. To fully access the possibility of GLUT1 and STn expressions in the same types of cells, we also performed double staining immunofluorescence for GLUT1 and STn in the same sections (data not shown because it retrieved negative). Again, we could not find any evidence of GLUT1-STn glycoproteoforms, reinforcing its highly specific cancer-associated nature. Nevertheless, at this stage, the existence of GLUT1-STn glycoproteoforms in the kidney remains a possibility, which warrants confirmation foreseeing targeted therapeutics. Moreover, detailed glycoproteomics studies by mass spectrometry are also required to unequivocally validate these preliminary observations.

## 3. Material and Methods

### 3.1. Patient Samples and Ethics Statement

Sialyl-Tn and Glucose transporter type 1 (GLUT1) analysis was performed retrospectively in a series of 48 formalin-fixed paraffin-embedded (FFPE) ESCC tissues obtained from Portuguese Institute of Oncology of Porto (IPO-Porto) biobank. Esophageal squamous cell carcinomas were surgically removed from 38 men and 10 women, ranging from 44 to 83 years of age (median 63 ± 11 years), admitted and treated at IPO-Porto between 2010 and 2015 (Table 1). Overall survival (OS) was defined as the period between surgical treatment and patients’ death by cancer. Recurrence was defined as the appearance of disease, locally or distant, after the first treatment. Clinicopathological information was obtained from patients’ clinical records, upon IPO institutional Ethics Committee approval (202/017 approved on 20 July 2017, addendum 202a/017 approved on 9 November 2017) and after patient’s informed consent. A broad library of healthy tissues (skin, thyroid, lung, stomach, liver, pancreas, gallbladder, colon, kidney, testis) was also screened for these markers.

### 3.2. Isolation and Characterization of Circulating Tumor Cells

The isolation and characterization of the STn antigen in EC-derived circulating tumor cells (CTCs) were performed prospectively in 10 male ESCC patients (age of diagnosis 63 ± 11 years). This corresponds to a group of patients admitted at IPO-Porto and enrolled in the study after informed consent and before initiation of any therapeutic procedure. The clinicopathological nature of the tumors is summarized in Table 2. CTCs were enriched from peripheral blood using a size-based microfluidics device developed at International Iberian Nanotechnology Laboratory (INL)-Braga and described in detail elsewhere [22,54]. Captured cells were stained in situ for CD45 (R&D Systems, Minneapolis, MN, USA), pan-Cytokeratin (pan-CK; Novus Biologicals, Centennial, CO, USA) and STn (B72.3+CC49; Abcam, Cambridge, UK) by immunofluorescence using a Leica DMI6000 FFW microscope (Leica Microsystems, Wetzlar, Germany) equipped with the Las X software (Leica Microsystems, Wetzlar, Germany). This enabled the estimation of STn positive cancer cells (DAPI^+^/CD45^−^/pan-CK^+^ or pan-CK^−^/STn^+^) as well as the assessment of hematopoietic contamination (DAPI^+^/CD45^+^/pan-CK^−^/STn^−^). Spiking experiments were performed using the MCRSTn^+^ bladder cancer cell line as a positive control of epithelial cancer cells expressing the STn antigen and CKs but lacking CD45 expression [24]. Blood cells from healthy donors were used as negative controls, containing cells positive for CD45 and negative for STn (DAPI^+^/CD45^+^/pan-CK^−^/STn^−^). In parallel, we incubated microchips containing isolated CTCs with neuraminidase (NeuAse) from *Clostridium perfringens* (Sigma-Aldrich, St. Louis, MO, USA) prior incubation with primary antibodies, as a negative control for STn. This was done based on a protocol optimized by us [22].

### 3.3. Cell Lines and Culture Conditions

The Human ESCC cell line Kyse-30 was kindly provided by Prof. Fátima Baltazar (Life and Health Sciences Research Institute from the University of Minho, Braga, Portugal). Upon arrival, cell line authentication by autosomal short tandem repeat (STR) DNA Profiling was performed [55] and the DSMZ STR profile database was used to query STR profiles, confirming its origin. Cells were maintained with 10% heat-inactivated fetal bovine serum (FBS) (Thermo Fisher Scientific, Waltham, MA, USA) and 1% penicillin-streptomycin (10,000 Units/mL penicillin; 10,000 mg/mL streptomycin; Thermo Fischer Scientific, Waltham, MA, USA) supplemented RPMI 1640+GlutaMAX^TM-I^ medium (Thermo Fisher Scientific, Waltham, MA, USA). The cell line was cultured at 37 °C in a 5% CO_2_ humidified atmosphere using a BINDER C-150 incubator (BINDER GmbH, Tuttlingen, Germany).

### 3.4. Generation of an STn ESCC Cell Line

Kyse-30 cells were plated onto 24-well plates to be 70% confluent at the time of transfection. Human *ST6GALNAC1* (h*ST6GALNAC1*, NM_018414.5) knock-in (KI) was achieved by conventional mammalian gene expression vector transfection using jetPRIME^®^ transfection reagent (Polyplus-transfection, Illkirch, France), according to the manufacturer’s instructions. In parallel, a mock system containing a 300 bp stuffer ORF in an empty vector was established. The KI system was purified through puromycin selection of positively transfected cells. Cells were subsequently sorted in a BD FACSAria™ III sorter for selecting high level STn expressing cells. Cell sorting was preceded by cellular detachment using Versene solution (Thermo Fisher Scientific, Waltham, MA, USA) and staining with mouse anti-TAG72 antibody [B72.3] (ab691; Abcam, Cambridge, UK) at 2 µg/10^6^ cells dilution in PBS 2% FBS for 1 h at room temperature. Polyclonal rabbit anti-mouse immunoglobulins/FITC (F0313; Agilent, Santa Clara, CA, USA) was used as secondary antibody for STn detection at a 1:100 dilution in phosphate buffered saline (PBS) 2% FBS for 15 min at room temperature.

### 3.5. Proliferation Assays

Cell proliferation was evaluated using a colorimetric immunoassay that measures the incorporation of 5-bromo-2′-deoxyuridine (BrdU) during DNA synthesis. Briefly 5 × 10^3^ cells/well were cultured into 96 well plates and the Cell Proliferation ELISA, BrdU (colorimetric) Kit (Roche Diagnostics GmbH, Mannheim, Germany) was used for the quantification of cell proliferation, according to the manufacturer’s instructions. Measurements of samples absorbances were taken at 450 nm using the iMARK^TM^ microplate reader (Bio-Rad, Hercules, CA, USA). Cell death negative controls composed of 1% Triton-X in complete cell culture medium were used in all experiments. All experiments were performed in triplicate and three replicates were conducted for each independent experiment. The results are presented as the average and standard deviation of these assays.

### 3.6. Invasion Assays

Corning BioCoat Matrigel Invasion Chambers (Corning, Corning, NY, USA) were used to assess the invasive properties of Kyse-30, Kyse-30 Mock and Kyse-30 *ST6GALNAC1* KI cells in vitro. Briefly, 5 × 10^4^ cells/mL were plated onto invasion inserts conditioned according to the manufacturer’s instructions. After 24 h assay, the membranes were washed and mounted with VECTASHIELD^®^ mounting medium with DAPI and cells were counted in a Leica DM2000 microscope (Leica Microsystems, Wetzlar, Germany). Three independent assays were performed and cells were seeded in quintuplicates for each cell line.

### 3.7. Flow Cytometry

Cells were detached using Versene solution (Thermo Fisher Scientific, Waltham, MA, USA), fixed with 2% paraformaldehyde (PFA; Sigma-Aldrich, St. Louis, MO, USA) and stained with mouse anti-TAG72 antibody [B72.3] (ab691; Abcam, Cambridge, UK) according with conditions previously described in *Generation of an STn ESCC cell line* topic. Mouse IgG1 [MOPC-21] isotype control (ab18443; Abcam, Cambridge, UK) was included as a negative control. In parallel, 10^6^ cells were digested with 70 mU neuraminidase from *Clostridium perfringens* (Sigma-Aldrich, St. Louis, MO, USA) in appropriate sodium acetate buffer at 37 °C overnight under mild agitation prior to STn staining, as a proper negative control. Data analysis was performed through CXP Software and results represent the standard deviation of three independent experiments performed in a FC500 Beckman Coulter flow cytometer (Beckman Coulter Life Sciences, Indianapolis, IN, USA).

### 3.8. Glycomics

ESCC cell models *O*-glycome were characterized using the Cellular *O*-glycome Reporter/Amplification method [25] and analyzed by nanoLC-ESI-MS/MS, as previously described by us [9]. Glycan structures were assigned based on *m*/*z* identification, retention time, characteristic product ion spectra and previous knowledge about *O*-glycosylation pathways [9]. Glycans were expressed in terms of relative abundance in comparison to the sum of all individual contributions to the glycome.

### 3.9. Immunohistochemistry

Three micrometers FFPE esophageal tumor sections were deparaffinized, rehydrated and subjected to antigen retrieval for 20 min with boiling citrate buffer for STn (Vector Laboratories, Burlingame, CA, USA) or 1 mM EDTA pH 9 for GLUT1 analysis. Endogenous peroxidases were inactivated by 3% hydrogen peroxide (Merck KGaA, Darmstadt, Germany) incubation for 5 min, followed by blockage of unspecific links for 5 min with Protein Block (Leica Biosystems, Wetzlar, Germany). Finally, tissue sections were separately incubated with 0.5 µg/mL monoclonal antibody (moAb) anti-STn (B72.3+CC49; Abcam, Cambridge, UK) and moAb anti-GLUT1 (EPR3915; Abcam, Cambridge, UK) at the dilution of 1:750. STn and GLUT1 were posteriorly detected using the Novolink Max Polymer DS Kit (Leica Biosystems, Wetzlar, Germany), according to the manufacturer’s instructions. Positive and negative controls were tested in parallel, including enzymatic controls with 0.2 U/mL α-neuraminidase from *Clostridium perfringens* (Sigma-Aldrich, St. Louis, MO, USA) overnight at 37 °C. Additionally, a broad panel of healthy tissues (skin, thyroid, lung, stomach, liver, pancreas, gallbladder, colon, kidney, testis) was also screened for both glycan and protein to assess the expression pattern in normal conditions. Cancer tissues were considered positive for STn whenever staining was observed, whereas the expression of GLUT1 was determined by multiplying the extension (0–100) against the intensity of expression (1: weak; 2: medium; 3: strong). Tumors showing GLUT1 levels above 200 were considered to overexpress this antigen. Healthy tissues were considered positive for the two antigens whenever staining was observed.

### 3.10. Double Staining Immunofluorescence Microscopy

STn and GLUT1 were evaluated in the same tumor sections by double staining immunofluorescence microscopy. Briefly, FFPE esophageal tumor sections were deparaffinized, rehydrated and incubated for 20 min with 1 mM EDTA pH 9 to recover STn and GLUT1 epitopes. Unspecific background was blocked with Protein Block (Leica Biosystems, Wetzlar, Germany) for 5 min and both markers (STn and GLUT1) were incubated with primary monoclonal antibodies, as previously described for immunohistochemistry staining. Posteriorly, moAb anti-STn and anti-GLUT1 were detected using an Alexa Fluor 488 goat anti-mouse IgG (Thermo Fisher Scientific, Waltham, MA, USA) and an Alexa Fluor 594 goat anti-rabbit IgG (H+L), respectively, at the dilution of 1:100 for 30 min at room temperature. Nuclear counterstaining was obtained using a 4′,6-diamidino-2-phenylindole, dihydrochloride (DAPI; Thermo Fisher Scientific, Waltham, MA, USA) solution for 10 min. Fluorescence images were acquired on a Leica DMI6000 FFW microscope (Leica Microsystems, Wetzlar, Germany) using the Las X software (Leica Microsystems, Wetzlar, Germany).

### 3.11. Immunoprecipitation and Western Blot

Proteins were extracted from 10 µm FFPE ESCC tissues using Qproteome FFPE tissue kit (Qiagen, Hilden, Germany), according to the manufacturer’s instructions. Prior to quantification of total protein extracts using a DC Protein Assay kit (Bio-Rad, Hercules, CA, USA), the elution buffer was exchanged to a Lysis buffer (50 mM Tris, 150 mM NaCl, 1% NP-40, 0.5% sodium deoxycholate, 0.1% SDS). For each immunoprecipitation (IP) reaction, Pierce^TM^ Protein G Agarose (Thermo Fisher Scientific, Waltham, MA, USA) was blocked with 1% Bovine Serum Albumin (BSA; Sigma-Aldrich, ST. Louis, MO, USA) for 1 h at 4 °C. Then, 100 µg whole cell protein extracts were pre-incubated with agarose beads blocked with BSA for 2 h at 4 °C to remove proteins showing unspecific binding to the columns. The protein extracts were then recovered and subsequently incubated with 3 µg of moAb anti-GLUT1 (EPR3915; Abcam, Cambridge, UK) at 4 °C for 2 h. BSA blocked agarose beads were then added to the medium and allowed to incubate at 4 °C overnight under gentle stirring. After five washing steps with the buffer provided by the vendor, the IP products were eluted from beads using an acidic IP elution buffer (0.1 M citrate pH 3). IPs controls were conducted in parallel using a rabbit IgG isotype control (02-6102; Thermo Fisher Scientific, Waltham, MA, USA). Finally, the IP products were neutralized, run on 4–20% gradient SDS-PAGE gels (Bio-Rad, Hercules, CA, USA), transferred into nitrocellulose membranes (GE Healthcare Life Sciences, Chicago, IL, USA) and screened for STn and GLUT1 antigens by Western Blot. Briefly, membranes were blocked with 1% Carbo-Free Blocking Solution (Vector Laboratories, Burlingame, CA, USA) and probed with each moAb, mentioned formerly, for 1 h at room temperature. MoAb anti-STn detection was performed using a peroxidase affiniPure goat anti-mouse IgG (H+L) polyclonal antibody, while moAb anti-GLUT1 was conjugated with a goat anti-rabbit IgG (H+L) HPR antibody for 30 min at room temperature. Finally, the Amersham ECL Prime Western Blotting Detection Reagent (GE Healthcare Life Sciences, Chicago, IL, USA) was used as developing reagent. Chemiluminescence detection was performed in a ChemiDoc XRS+ with Image Lab^TM^ Software (Bio-Rad, Hercules, CA, USA).

### 3.12. Bioinformatics for Biomarker Discovery

STn bearing glycoprotein discovery was performed exploiting the only available proteomics dataset for ESCC (project PXD006255; https://www.ebi.ac.uk/pride/archive/projects/PXD006255 (accessed on 1 January 2021)) deposited in the EMBL-EBI proteomics identification database (PRIDE; https://www.ebi.ac.uk/pride/ (accessed on 1 January 2021)) until April 2020. The dataset reports data previously published by Puttamallesh et al. [56] and consists of the nanoLC-MS analysis of a combined protein pool extracted from 10 µm FFPE tissue sections from 6 ESCC patients by high resolution tandem mass spectrometry. The dataset comprises nanoLC-HCD-MS/MS data generated on an Orbitrap Fusion™ Tribrid™ Mass Spectrometer for the analysis of tryptic digests after previous fractionation by strong cation exchange (SCX) or reverse phase sulfonate (RPS) chromatography. Data were re-analyzed using the SequestHT search engine with the Percolator algorithm for validation of protein identifications (Proteome Discoverer 2.5; Thermo Fisher Scientific, Waltham, MA, USA). A double search strategy was devised first aiming at identifying relevant cell membrane proteins and secondly trying to map the proteins *O*-glycosites. Initially, data were searched against the human plasma membrane proteome obtained from the SwissProt database in March 2020. Trypsin was selected as the enzyme and up to 2 missed cleavages were allowed. Precursor ion mass tolerance was set at 10 ppm and 0.02 Da for product ions. Carbamidomethylcysteine was selected as a fixed modification, while oxidation of methionine (+15.9949) was defined as variable modification. High confidence identifications were ordered in relation to their protein scores and the top 250 membrane proteins from the SCX and RPS pre-fractionation were combined for common molecular signatures. The resulting list was re-organized using a Target Score, as previously described in detail by us [9]. The Target Score algorithm was designed to attribute higher scores to proteins overexpressed in cancer, located at the cell membrane and associated with poor prognosis, while penalizing proteins observed at the same cellular location in healthy tissues. As primary source of information for differently expressed proteins, we used the Human Protein Atlas database (https://www.proteinatlas.org/ (accessed on 1 January 2021)) consulted in March 2020 [57]. More details on the algorithm are presented as Appendix A. Since the Human Protein Atlas at the time of consultation presented no information concerning ESCC, the present study adopted protein expression levels in head-and-neck squamous cell carcinoma as a model, based on exposure to the same carcinogens, similar histopathology and molecular traits [33,34,35]. For the second search, the top 25 Target Score ranked proteins were constituted as a customized database and interrogated for STn-glycosites using the Byonic software v3.10-52 (Protein Metrics, Cupertino, CA, USA) [41]. Briefly, precursor ion mass tolerance was set at 10 ppm and 0.01 Da for product ions. Carbamidomethylcysteine was selected as a fixed modification, while oxidation of methionine (+15.9949 Da) and the presence of the STn antigen (+494.1748 Da) were considered as variable modifications.

### 3.13. Statistical Analysis

Statistical data analysis was performed with IBM Statistical Package for Social Sciences—SPSS for Windows (version 20.0, IBM, Armonk, NY, USA). Non-parametric Mann-Whitney test was used for proliferation and invasion assays. Chi-square analysis was used to compare categorical variables. Kaplan-Meier curves were used to analyze the influence of the biomarkers expression in the context of time-to-event (death). Multiple Cox regression analysis was used to assess the effect of the studied biomarkers on the time to recurrence and to adjust for potentially confounding effects. Comparison of estimates was done using log-rank test. A 95% significance threshold for the null hypothesis was considered.

## 4. Conclusions

ESCC remains a challenging disease due to the lack of cancer-specific cell surface biomarkers to aid therapeutic decision and precisely target cancer cells. Herein, we have demonstrated that more aggressive tumors facing worst prognosis overexpress the STn antigen, potentially mirroring profound de-regulation in protein *O*-glycosylation pathways yet to be fully uncovered for these tumors. High STn expression was associated with increased probability of developing distant metastases, which was supported by the identification of STn in CTCs and reinforced by studies in vitro linking STn to increased cellular invasion. Notably, this is also the first report of STn in esophageal CTCs, which follows similar observations for bladder, colorectal and gastric tumors [21,22]. Future studies must assess its potential clinical value in the context of liquid biopsies, namely for early detection of disseminated disease, prognosis, treatment follow-up and other molecular-based clinical decisions. Nevertheless, we must emphasize the exploratory nature of this study, involving a low number of tumor tissues and blood samples, warranting confirmation in larger patient cohorts. Notably, we have also observed low amounts of STn in histologically normal esophagus and healthy cells of the gastrointestinal and respiratory tracts. These findings prompted an investigation of the ESCC STn-glycoproteome envisaging specific glycosignatures that could overcome limitations associated with possible off-target effects. As proof-of-concept we have applied the Target Score algorithm to pinpoint potentially clinically relevant glycoproteins identified from large proteomics datasets. Faced with absence of information on protein expression for EC, we used as primary source of information squamous cells head-and-neck tumors, which share similar etiology, clinicopathological and molecular natures. Even though of preliminary nature and strongly depending on the quality of the initial database, this approach led to the identification of GLUT1 as a cancer-associated differentially expressed protein. Building on these findings, we have demonstrated that GLUT1 was modified with STn antigens. We also showed that GLUT1-STn overexpression may hold potential for ESCC patient stratification, enabling the identification of subgroups facing worst prognosis, as previously suggested when the two antigens were evaluated separately. These findings now warrant definitive confirmation in a broader patient series including different clinicopathological natures and detailed clinical histories, which is currently ongoing. Furthermore, we are addressing the functional implications of GLUT1-STn molecular signature for ESCC to help shaping the necessary rationale for biomarker interventions. Foreseeing this objective, we have also conducted an exploratory screening of relevant healthy tissues of distinct systems (reproductive, respiratory, digestive, immune), which suggested that GLUT1-STn glycoforms may be mainly present in cancer. Nevertheless, our studies were based on immunoassays that strongly depend on the specificity of the antibodies used for detection. As such, a more comprehensive interrogation of the human proteome using high throughput mass spectrometry is warranted to support this claim. Finally, our investigation supports previous reports from our group highlighting the clinical relevance of specific protein glycosignatures, namely for improved patient stratification and increased cancer specificity. Yet, we reinforce that this was a proof-of-concept study exploiting a pre-existing proteomics data generated from a pool of patients without a defined clinical characterization. The original study was not initially designed to accommodate the subtilities of glycoproteomics analysis, including prior enrichment of the samples for glycoproteins and mass spectrometry methodologies to support precise glycosites assignment and higher protein coverages. Nevertheless, it was sufficient to provide molecular signatures of interest, setting a rationale that may be generalized to other tumors. Moreover, it strongly encourages a revisitation of ESCC with dedicated glycoproteomics protocols towards true precision medicine settings based on glycobiomarkers.

## Figures and Tables

**Figure 1 ijms-22-01664-f001:**
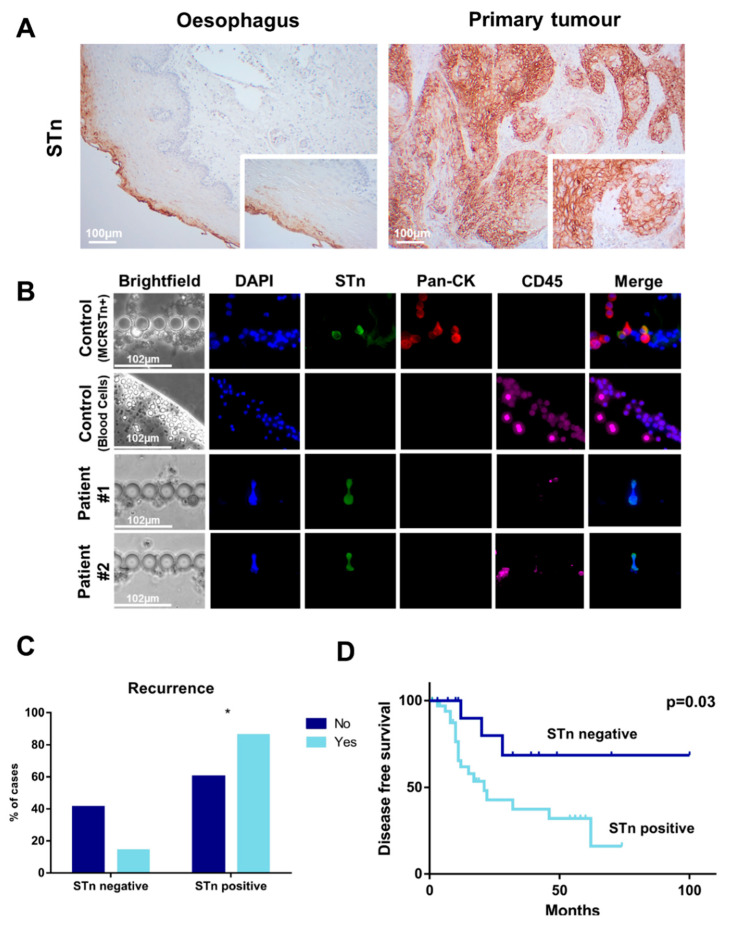
The STn antigen is overexpressed in esophageal squamous cell carcinomas (ESCC) and CTCs, being associated with recurrence and decreased disease-free survival. (**A**) The STn antigen is overexpressed in approximately 70% of ESCC and adjacent lymph node metastasis and shows little expression in the healthy esophagus. Most ESCC and lymph node metastasis present focal STn expressions showing high intensity staining at the cell membrane. On the other hand, the healthy esophagus shows a diffuse low to moderate cytoplasmatic staining in differentiating cells of the mucosa facing the lumen of the organ. (**B**) CTCs recovered from the blood of ESCC patients express the STn antigen. CTCs were isolated from whole blood by size-based microfluidics devices presenting an internal architecture that enables the entrapment of larger cancer cells with minimal contamination from blood cells. In addition to STn (green), cell nuclei were stained with 4′,6-diamidino-2-phenylindole, dihydrochloride (DAPI) (blue), pan-CKs (red) was used as an epithelial marker, generally expressed by some subpopulations of cancer cells and CD45 (purple) was used to identify blood cells. ESCC-derived CTCs presented a DAPI^+^/pan-CKs^-^/STn^+^/CD45^−^ phenotype. MCRSTn^+^ bladder cancer cells were used as positive control for cancer cells expressing the STn antigen (DAPI^+^/pan-CKs^+^/STn^+^/CD45^−^) and blood cells from healthy donors were used to demonstrate the absence of STn in CD45^+^ cells (DAPI^+^/pan-CKs^−^/STn^−^/CD45^−^). (**C**) STn expression associates with recurrence (* *p* < 0.05; Chi-square) and (**D**) disease-free survival (*p* = 0.03, Chi-square).

**Figure 2 ijms-22-01664-f002:**
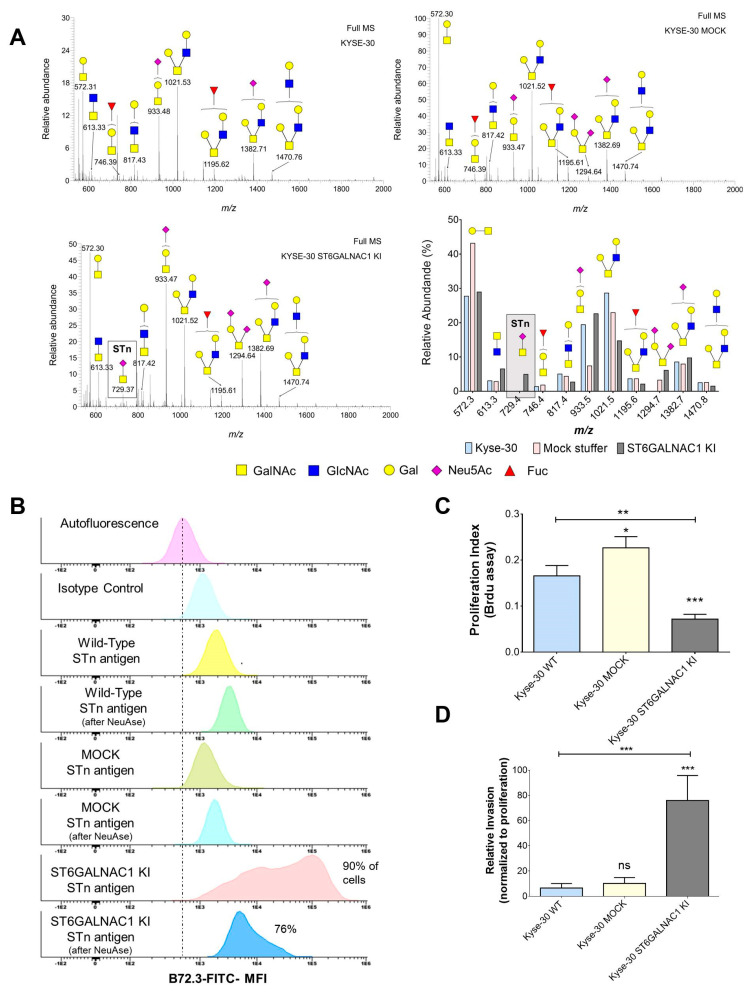
The stable transfection of *ST6GALNAC1* leads to STn expression in Kyse-30 cells without inducing profound glycome remodeling and induces a massive decrease in proliferation and a significant increase in invasion. (**A**) Kyse-30 cells express elongated *O*-glycans and do not express STn, whereas Kyse-30 *ST6GalNAc1* KI express STn with minor alterations in the glycome. Glycomics analysis on Kyse-30 revealed a glycome dominated by several core 2 *O*-glycans (*m*/*z* 817.43; 1021.53; 1195.62; 1382.71; 1470.76) as well as T (*m*/*z* 572.31) and sialylated T antigens (*m*/*z* 933.48). Core 3 was also detected. (*m*/*z* 613.33). Mock cells presented a similar profile, even though showing the T antigen (*m*/*z* 572.3) as major ion and presenting low amounts of di-sialyl-T (*m*/*z* 1294.7). Neither wild type nor mock cells expressed the STn antigen (*m*/*z* 729.4). On the other hand, *ST6GALNAC1* transfected cells presented a glycome very similar to wild type cells but were the only cells expressing STn (*m*/*z* 729.4). (**B**) Flow cytometry analysis confirms the low STn levels in Kyse-30 and the massive increase in this antigen after stable transfection with *ST6GalNAc1*. The stable transfection with *ST6GALNAC1* leads to a significant increase in STn in almost 90% of the cells. The signal significantly decreases with the removal of the sialic acid after sialidase (NeuAse) digestion, confirming the presence of STn. Notably, the extension of NeuAse digestions is never complete, explaining the presence of lower amounts of STn in the cells after treatment. (**C**) The overexpression of STn resulting from ST6GalNAc I overexpression leads to a massive decrease in proliferation. (**D**) The overexpression of STn resulting from ST6GalNAc I overexpression leads to a massive increase in invasion. ns: not significative; * *p* < 0.05; ** *p* < 0.01; *** *p* < 0.001; Mann-Whitney for three independent replicates in proliferation assays and five independent replicates in invasion assays.

**Figure 3 ijms-22-01664-f003:**
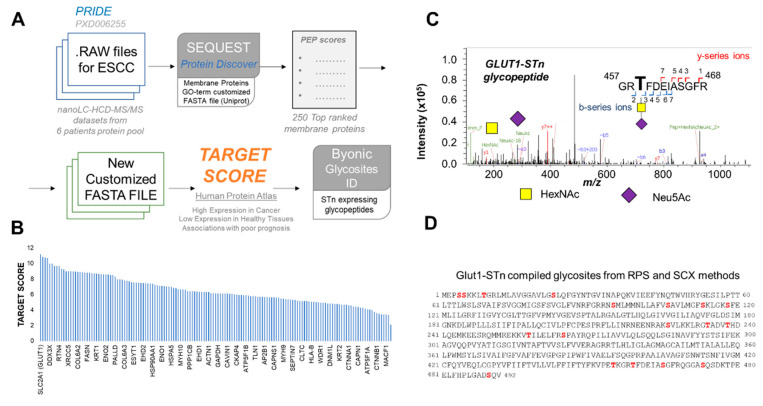
Target score assisted revisitation of ESCC proteomics data identifies SLC2A1 (GLUT1) as a potentially targetable biomarker carrying the STn antigen. (**A**) Bioinformatics workflow for identification of targetable biomarkers in ESCC. Raw ESCC nanoLC-MS/MS proteomics data associated with project PXD006255 were first downloaded from the PRIDE database and screened for membrane glycoproteins using SEQUEST, a tandem mass spectrometry database search program for Proteome Discoverer. The original dataset comprehended nanoLC-MS/MS runs generated from protein digests with Lys-C and trypsin. The proteins were isolated from a pool of 6 ESCC. Two-types of pre-separation methods were employed prior to nanoLC-MS/MS analysis: reverse phase sulfonate (RPS) and strong cation exchange (SCX) methods. As such, LC-MS/MS data from the two methods were processed separately. Identifications were made against a Gene Ontology (GO) term customized database for plasma membrane proteins and ordered in relation to their posterior error probability (PEP) score. Top ranked 250 proteins for each method were then matched to find common signatures that were subsequently analyzed by the Target Score algorithm. Target score provided an easy strategy for protein classification according to potentially targetability based on pre-existing information about cellular location, expression in tumors and human healthy tissues as well as associations with poor prognosis. The algorithm uses as primary source of information the Human Protein Atlas but the same workflow may be applied to similar databases. Finally, we returned to the initial proteomics raw data to access if top scored glycoproteins were modified with the STn antigen, using the Byonic software for more precise glycopeptides identification and glycosites annotation. Collectively, this strategy was designed to identify with high confidence abnormally glycosylated proteins with the STn antigen, showing higher prevalence at the cell membrane of ESCC in comparison to healthy tissues, enabling easy targeting with antibodies and other ligands with minimal potential off-target effects. (**B**) Target Score analysis identified SLC2A1 (GLUT1) as a potential targetable molecule in ESCC. (**C**) Example of a glycopeptide high-energy collision dissociation-tandem mass spectrometry (HCD-MS/MS) spectrum confirming the existence of GLUT1-STn glycoproteoforms in ESCC. The MS/MS spectrum highlights characteristic b- and y-series ions derived from fragmentations in the peptide backbone, enabling glycosites assignment, as well as GalNAc and Neu5Ac oxonium ions from STn. (**D**) STn expressing glycosites in GLUT1. STn glycosites in GLUT1 are highlighted in red. Details on the identified glycopeptides are presented as Appendix A.

**Figure 4 ijms-22-01664-f004:**
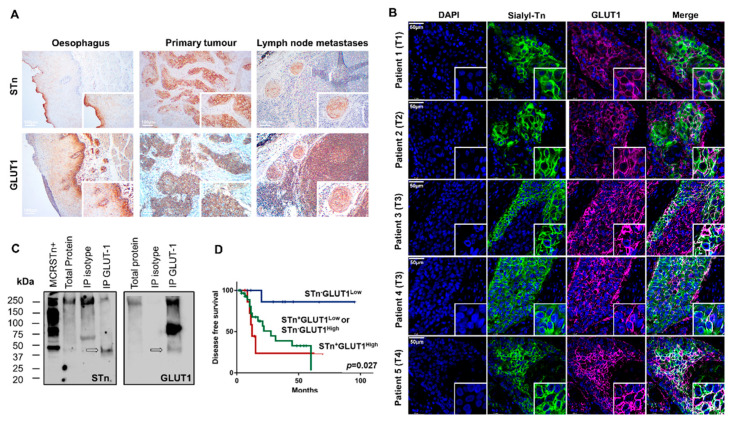
GLUT1-STn is widely expressed in ESCC and correspondent metastases and associates with decreased disease-free survival. (**A**) GLUT1 and STn are co-localized in the same tumor areas in primary tumors, lymph node metastases but not in the healthy esophagus. The immunohistochemistry analysis panels show the co-expression of GLUT1 and STn in the same areas in tumors and lymph node metastases strongly suggesting GLUT1-STn glycoforms. Immunohistochemistry of the healthy esophagus suggests that these epitopes are present in different areas. (**B**) GLUT1 and STn are expressed in the same cancer cells in all studied tumors. Double immunofluorescence for GLUT1 and STn show co-localization in the same cells in all studied tumors, across different stages and histopathological natures of the disease. Nuclei stained in blue; STn in green; GLUT1 in purple; Co-expression (white). (**C**) Western blot confirms the existence of GLUT1-STn glycoproteoforms. GLUT1 was immunoprecipitated from ESCC and blotted for both GLUT1 and STn. Isotype immunoprecipitated proteins were used as negative controls. MCRSTn^+^ bladder cancer cell line was used as positive control for STn expression. All blots showed bands at 250 kDa and above most likely resulting from unspecific co-precipitation of mucins carrying STn. On the other hand, GLUT1 blots evidence a major band just above 75 kDa not present in STn blots. A less intense band at approximately 50 kDa was observed in both GLUT1 and STn blots reinforcing the existence of GLUT1-STn. (**D**) GLUT1-STn overexpressing tumors present reduced decreased survival when compared to tumors showing no STn expression or high STn expression and decreased levels of GLUT1.

**Figure 5 ijms-22-01664-f005:**
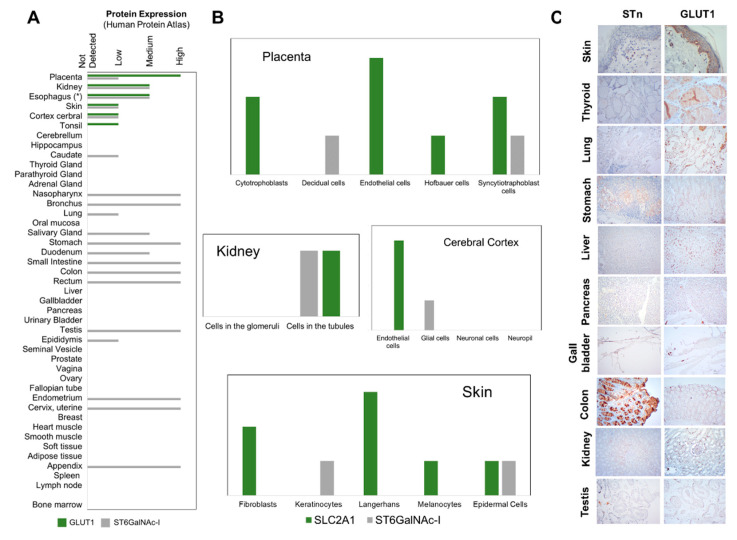
STn and GLUT1 expressions are rarely overlapping in healthy tissues. (**A**) GLUT1 and ST6GalNAc I (a key glycosyltransferase for STn biosynthesis) expressions in healthy human tissues. According to the Human Protein Atlas, GLUT1 and ST6GalNAc I are co-expressed in the placenta, kidney, skin and the cerebral cortex, suggesting potential to express GLUT1-STn. GLUT1 for the esophagus concerned experimental data from the present report, according to Figure 4A. (**B**) GLUT1 and ST6GalNAc I expressions in different cell types of the placenta, kidney, cerebral cortex and skin. Syncytiotraphoblast cells of the placenta, cells in the kidney tubules and epidermal cells of the skin co-express GLUT1 and ST6GalNAc I, supporting capacity to present GLUT1-STn. (**C**) GLUT1 and STn expressions in the skin, thyroid, lung, stomach, liver, pancreas, gallbladder, colon, kidney, testis. GLUT1 was detected at the cell membrane in the immune cells’ component of the stomach, liver, colon, lung and pancreas. STn was found mostly in the cytoplasm and, to less extent, the membrane of parietal and goblet cells of the respiratory and digestive tracts. No evidence supporting co-expression in the same areas were found.

**Table 1 ijms-22-01664-t001:** Clinicopathological data of patients included in the sialyl-Tn (STn) analysis (*n* = 48).

	*n* (%)
**Stage**	
I	13 (27)
II	11 (23)
III	23 (48)
IV	1 (2)
**Tumour (pT)**	
T1	9 (19)
T2	13 (27)
T3	25 (52)
T4	1 (2)
**Lymph node metastasis (pN)**	
N0	21 (44)
N1	11 (23)
N2	10 (21)
N3	6 (12)
**Distant Metastasis (M)**	
M0	47 (98)
M1	1 (2)
**Distant Recurrence (DR)**	
No	31 (65)
Yes	17 (35)
**Histological Classification**	
Squamous cell carcinoma	48 (100)
Adenocarcinoma	0 (0)
**Keratinization Degree**	
Non-keratinized	30 (63)
Moderately keratinized	2 (4)
Keratinized	16 (33)
**Differentiation Degree**	
Well-differentiated	7 (15)
Moderately differentiated	26 (54)
Poorly differentiated	10 (21)
Missing information	5 (10)
**Extra-tumoral growth**	
Present	12 (25)
Absent	36 (75)
**Lymphovascular Permeation**	
Present	21 (44)
Absent	27 (56)
**Neural Permeation**	
Present	13 (27)
Absent	35 (73)

**Table 2 ijms-22-01664-t002:** Clinicopathological data of patients included in circulating tumor cells (CTCs) analysis (*n* = 10).

	*n* (%)
**Stage**	
I	0 (0)
II	7 (70)
III	3 (30)
IV	0 (0)
**Tumour (pT)**	
T1	1 (10)
T2	2 (20)
T3	6 (60)
T4	1 (10)
**Lymph node metastasis (pN)**	
N0	7 (70)
N1	3 (30)
**Distant metastasis (M)**	
M0	10 (100)
M1	0 (0)
**Histological Classification**	
Squamous cell carcinoma	10 (100)
Adenocarcinoma	0 (0)
**Lymphovascular Permeation**	
Present	3 (30)
Absent	3 (30)
Missing information	4 (40)
**Neural Permeation**	
Present	1 (10)
Absent	5 (50)
Missing information	4 (40)

**Table 3 ijms-22-01664-t003:** STn expression in esophageal cancer (EC) according to clinicopathological variables.

STn Expression	Positive Cases/Total (%)	*p* Value
**Stage**		
I	7/13 (54)	0.414
II	8/11 (72)	
III	18/23 (78)	
IV	1/1 (100)	
**Tumour (pT)**		
T1	4/9 (44)	0.261
T2	10/13 (77)	
T3	19/25 (76)	
T4	1/1 (100)	
**Lymph node metastasis (pN)**		
N0	13/21 (62)	0.639
N1	8/11 (73)	
N2	8/10 (80)	
N3	5/6 (83)	
**Distant metastasis (M)**		
M0	33/47 (70)	0.708
M1	1/1 (100)	
**Distant recurrence (DR)**		
No	20/31 (65)	0.167
Yes	14/17 (82)	
**Borrmann Classification**		
I	4/7 (57)	0.583
II	15/21 (71)	
III	6/7 (86)	
IV	9/11 (82)	
Missing information	0/2 (0)	
**Keratinization Degree**		
Non-keratinized	22/30 (73)	0.762
Moderately keratinized	1/2 (50)	
Keratinized	11/16 (69)	
**Differentiation Degree**		
Well-differentiated	4/7 (57)	0.577
Moderately differentiated	19/26 (73)	
Poorly differentiated	8/10 (80)	
Missing information	3/5 (60)	
**Extra-tumoral growth**		
Present	10/12 (83)	0.237
Absent	24/36 (67)	
**Lymphatic Permeation**		
Present	16/21 (76)	0.347
Absent	18/27 (67)	
**Vascular Permeation**		
Present	14/21 (67)	0.403
Absent	20/27 (74)	
**Neural Permeation**		
Present	8/13 (62)	0.301
Absent	26/35 (74)	

**Table 4 ijms-22-01664-t004:** GLUT1 expression in EC according to clinicopathological variables.

GLUT1 Expression	Low Expression (% Total)	High Expression (% Total)	*p* Value
**Stage**			
I	8 (62)	5 (38)	0.081
II	10 (91)	1 (9)	
III	19 (83)	4 (17)	
IV	0 (0)	1 (100)	
**Tumour (pT)**			
T1	8 (89)	1 (11)	0.406
T2	8 (62)	5 (38)	
T3	20 (80)	5 (20)	
T4	1 (100)	0 (0)	
**Lymph node metastasis (pN)**			
N0	15 (71)	6 (29)	0.575
N1	10 (91)	1 (9)	
N2	7 (79)	3 (30)	
N3	5 (83)	1 (17)	
**Distant metastasis (M)**			
M0	37 (79)	10 (21)	0.064
M1	0 (0)	1 (100)	
**Distant recurrence (DR)**			
No	27 (87)	4 (13)	**0.026**
Yes	10 (59)	7 (41)	
**Keratinization Degree**			
Non-keratinized	23 (77)	7 (23)	0.727
Moderately keratinized	2 (100)	0 (0)	
Keratinized	12 (75)	4 (25)	
**Differentiation Degree**			
Well-differentiated	3 (43)	4 (57)	0.081
Moderately differentiated	20 (77)	6 (23)	
Poorly differentiated	9 (90)	1 (10)	
Missing information	5 (100)	0 (0)	
**Extra-tumoral growth**			
Present	9 (75)	3 (25)	0.563
Absent	28 (78)	8 (22)	
**Lymphatic Permeation**			
Present	15 (71)	6 (29)	0.316
Absent	22 (81)	5 (19)	
**Vascular Permeation**			
Present	15 (71)	6 (29)	0.316
Absent	22 (81)	5 (19)	
**Neural Permeation**			
Present	10 (77)	3 (23)	0.632
Absent	27 (77)	8 (23)	

## Data Availability

Data is contained within the article or Appendix A.

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
