# Peer review of "Target Score—A Proteomics Data Selection Tool Applied to Esophageal Cancer Identifies GLUT1-Sialyl Tn Glycoforms as Biomarkers of Cancer Aggressiveness"

_ijms, 2021, doi:10.3390/ijms22041664_

Round 1
Reviewer 1 Report
Cotton etc al., has presented a very detailed Mass-spec proteomics analysis of Esophageal Squamous cells carcinoma and presented their argument of post-translational modifications of Sialytion / glycosylation of proteins as potential biomarker for improved and potentialy early detection of tumour. Authors have shown in tumour biopsies expression of sialyl-nT antigen as specifically expressed in tumour tissues. Modification of proteins is confirmed immunologically and biochemmically and convincingly shownn in the manuscript. Although this modification has been described before in literature and has been studied in other carcinomas, authors have shown it in Esophageal tumours and hence add the value to previous studies in addition to showing new data set of mass-spec proteomics. Authors have shown disease free survival of patient is significantly when patient do not express sTn on the tumour cells, though it is statistically significant but the number of patient is too small. This is an inherent issue with such specific studies. Authors are recommended to highlight this in their discussion as in current form it is overlooked. As shown in figure 2A relative abundance of STn is about 10%, it does look like authors are over stating this in the manuscript. Having said that in that very figure immunofluorescence data is supporting the hypothesis of authors. Figure 4C right panel shows a significant band at 75kDa, which appears to be unique in this IP, authors are not explaining why and how that band is appearing. Authors are arguing that STn is normally expressed in muucin secreting tissue, however, their own figure 5C does show expressing of STn in normal stomach and in testicular tissue. Authors should reword or discuss that expression in their manuscript. Overall, manuscript should benefit from editing as it will improve format improvements / font mismatch and with some typographical mistakes.
Reviewer 2 Report
- In Fig. 1C, 2 out of 10 CTCSs are STn-positive. The sample size is too few, although the authors mentioned this deficit in the text. This data may not be included in this study.
- STn over-expression is not related to all the clinicopathological parameters. However, the authors found that it is linked to recurrence. How do the authors determine the recurrence?
- The authors implement the first stage of bioinformatics to find 213 glycoproteins. However, they can not proceed to the next stage because of the lack of ESCC human Atlas. Thus they carried out the complete analysis of head-and-neck tumors. The authors should show the data of head-and-neck instead of ESCC.
- In Fig 4C., GLUT1’s molecular weight is 55 kDa. There is no 55 kDa protein band in GLUT1 blot, but the authors indicate that a protein less than 50 is STn-GLUT1. Also, there is no 75- and 55- KDa protein bands in total protein control in the GLUT1 blot. The authors may explain the above address.
- I can not understand why the authors include the expression of GLUT1 and STn in normal tissues. The expression of these two proteins almost are produced in different tissues.
Round 2
Reviewer 2 Report
I have no further comments.